# Seroprevalence and Molecular Epidemiology of Aleutian Disease in Various Countries during 1972–2021: A Review and Meta-Analysis

**DOI:** 10.3390/ani11102975

**Published:** 2021-10-15

**Authors:** Magdalena Zaleska-Wawro, Anna Szczerba-Turek, Wojciech Szweda, Jan Siemionek

**Affiliations:** 1District Veterinary Inspectorate in Olsztyn, Lubelska 16, 10-404 Olsztyn, Poland; magdalenawawro@wp.pl; 2Department of Epizootiology, Faculty of Veterinary Medicine, University of Warmia and Mazury in Olsztyn, Oczapowskiego 13, 10-718 Olsztyn, Poland; szweda@uwm.edu.pl (W.S.); jan.siemionek@uwm.edu.pl (J.S.); 3National Consultant for Diseases of Fur-Bearing Animals in Poland, Oczapowskiego 13, 10-718 Olsztyn, Poland

**Keywords:** *Parvoviridae*, epidemiology, Aleutian disease, Aleutian mink disease virus, *Neogale vison*, *Mustela lutreola*

## Abstract

**Simple Summary:**

Aleutian disease is caused by the Aleutian mink disease virus and is one of the most serious infectious diseases that affect the family Mustelidae, including the American mink, wild European mink, weasels, badgers and other animal species, such as skunks, raccoons, dogs, cats and mice, as well as humans. Effective treatments and vaccines against Aleutian disease have not been developed to date. Prophylactic programs that focus on the identification and elimination of infected mink are one of the methods of controlling the negative outcomes of Aleutian disease. This article analyses the seroprevalence of Aleutian mink disease virus infections in American and European mink and other species around the world, and reviews recent knowledge relating to the molecular epidemiology of the Aleutian mink disease virus.

**Abstract:**

Aleutian disease (AD) poses a serious threat to both free-ranging and farmed mink around the world. The disease is caused by the Aleutian mink disease virus (AMDV), which also poses a health risk for other members of the family Mustelidae, including wild mink, weasels, badgers and other animal species. This article analyses the seroprevalence of AMDV infections in mink and other species around the world, and reviews recent knowledge relating to the molecular epidemiology of the AMDV. Depending on the applied diagnostic technique and the country, the prevalence of anti-AMDV antibodies or AMDV DNA was established at 21.60–100.00% in farmed American mink, 0.00–93.30% in free-ranging American mink and 0.00–25.00% in European mink. Anti-AMDV antibodies or AMDV DNA were also detected in other free-living fur-bearing animals in Europe and Canada, where their prevalence was determined at 0.00–32.00% and 0.00–70.50%, respectively. This may indicate a potential threat to various animal species. AMDV strains are not clustered into genotypes based on the geographic origin, year of isolation or pathogenicity. The isolates that were identified on mink farms around the world originated from North America because American mink were introduced to Europe and Asia for breeding purposes and to restock natural populations.

## 1. Introduction

Aleutian disease (AD), also known as mink plasmacytosis or Aleutian mink disease (AMD), is one of the most serious infectious diseases that affect American mink (*Neogale vison*) and free-ranging European mink (*Mustela lutreola*) around the world [1]. AD leads to death and spontaneous abortion of mink, and it causes significant economic losses in the mink-farming industry [1,2,3]. The disease is caused by the Aleutian mink disease virus (AMDV), which also poses a health risk for other members of the family Mustelidae, including wild mink, weasels, badgers and other animal species, such as skunks, raccoons, dogs, cats and mice, as well as humans [1,4,5,6,7]. Infected mink with and without symptoms of the disease excrete viral particles with faeces into the environment. On American mink farms, the AMDV can be spread horizontally, mainly via blood, during sampling, faeces, urine and saliva, or vertically from mothers to the offspring. Infection occurs mainly via the gastrointestinal tract, but air-borne, iatrogenic and vector-borne infections were also reported [8,9,10]. The infection stimulates the production of anti-AMDV antibodies that are unable to neutralise the AMDV, which leads to hypergammaglobulinemia and the formation of autoimmune complexes. When these complexes are deposited in tissues, they cause inflammation and pathological changes that are accompanied by clinical symptoms, such as renal failure or respiratory distress [11]. Research indicates that AMDV strains are not grouped based on virulence, geographic location or the time of isolation [12]. The American mink widely colonised Canada and the USA. Since the 1950s, the species has also been encountered in the wild in Europe and Asia, where it had been introduced to the natural environment and imported for farming [13,14]. According to some researchers, American mink could pose a serious threat to the survival of native species [13,15,16]. Some scientists claim that American mink led to the decline of the European mink population, whereas others argue that it merely replaced European mink that had become extinct due to over-hunting [17].

For many years, AMDV was regarded as the sole member of the genus *Amdovirus* in the subfamily *Parvovirinae*, family *Parvoviridae*, but in 2014, it was reclassified as *Carnivore amdoparvovirus 1* of the genus *Amdoparvovirus*, subfamily *Parvovirinae* and family *Parvoviridae* [18]. The AMDV is a single-stranded DNA virus. The AMDV genome is approximately 4.8 kilobases long and consists of two capsid proteins, namely, VP1 and VP2, as well as three non-structural proteins, namely, NS1, NS2 and NS3 [7]. Capsid proteins determine the virus’s immunogenicity, the targeted animal species and viral tropism. VP2 is the main capsid protein, and its coding sequence contains a hypervariable region [19]. NS1 is essential for viral replication and is characterised by a high degree of genetic variability. The NS1 and VP2 genes have been widely used to assess the differences between AMDV strains worldwide, where epidemiological studies have shown that different AMDV strains vary considerably in terms of the genes encoding both NS and VP [20,21,22,23,24].

This article analyses the seroprevalence of AMDV infections in American and European mink and other species around the world, and reviews recent knowledge relating to the molecular epidemiology of the AMDV.

## 2. Materials and Methods

### 2.1. Selection of Scientific Papers

The PubMed engine (National Library of Medicine, 8600 Rockville Pike, Bethesda, MD 20894, USA, https://pubmed.ncbi.nlm.nih.gov/, accessed on 25 March 2021) was used to search for original scientific papers that were published in English between January 1972 and March 2021. The following keywords were used in the search: (AD* OR AMD*) AND (American mink* OR European mink* OR *Mustelides** OR fur-bearing animals* OR mink) AND (CIEP* OR ELISA* OR PCR) AND (Finland* OR Sweden* OR Estonia* OR Poland* OR Spain* OR Denmark* OR Canada* OR China*). These countries were chosen because mink farming is or had been an important part of their agricultural production. All retrieved records were saved for further review. Publications describing methods for diagnosing AD, determining the prevalence of anti-AMDV antibodies or AMDV DNA and molecular epidemiology data in the abstract were selected for preliminary analysis. Scientific papers dealing with diagnostic and molecular epidemiology, especially in European countries, were included. A total of 56 papers were analysed.

### 2.2. Data Extraction

From each selected publication, the following information was extracted and compiled in separate records (data points): (1) country of study, (2) types of animals, (3) sample size, (4) diagnostic and detection methods, (5) the prevalence of anti-AMDV antibodies or AMDV DNA and (6) molecular epidemiology.

## 3. AD Seroprevalence

Counter-current immunoelectrophoresis (CIEP) and enzyme-linked immunosorbent assay (ELISA) are the most popular techniques for analysing the seroprevalence of AMDV infections [25,26,27,28,29,30,31,32,33].

In studies that relied on the CIEP or ELISA techniques, the prevalence of anti-AMDV antibodies was determined to be in the range of 0.00 to 61.66% (arithmetic mean (A.M.) 32.07%) in free-ranging American minks [5,34], 23.80 to 82.60% (A.M. 59.89%) in farmed American minks [3,34,35,36,37] and 0.00 to 32.00% (A.M. 17.30%) in European minks [5,13,38]. The prevalence of anti-AMDV antibodies in different countries is shown in Table 1. The reported results are very difficult to interpret due to considerable differences in the sizes of the studied populations. 

In Canada, the prevalence of anti-AMDV antibodies differed significantly across studies and was determined to be in the range of 28.60 to 61.66% (A.M. 39.75%) in free-ranging American minks [4,34,39] and 23.80 to 70.70% (A.M. 46.94%) in farmed American minks [3,34]. Farid et al. [3] demonstrated that the CIEP-based test-and-removal strategy was effective in reducing the prevalence of AMDV-positive animals, but failed to eradicate the AMDV from infected farms [3]. In a CIEP analysis, specific anti-AMDV antibodies were identified in 51.85% of free-ranging American minks that were harvested in the Upper Thames Valley in southern England [15]. According to the researchers, the above could pose a serious threat to protected animal species, such as otter (*Lutra lutra*) and polecat (*Mustela putorius*) [15]. In Denmark, the prevalence of anti-AMDV antibodies reached 45.10% in Bornholm and 3.30% in the mainland [40]. A protection program was implemented in France to counteract the decline in the European mink population [41]. In 1996–2000, serum samples were collected from free-ranging European minks in southwest France. Specific anti-AMDV antibodies were detected in 12.12% of the tested animals (Table 1). Anti-AMDV antibodies were also identified in 11.00% of polecats (*Mustela putorius*), 23.50% of stone martens (*Martes foina*), 6.25% of pine martens (*Martes martes*) and 4.41% of common genets (*Genetta genetta*) [41]. In a study conducted by the same research team, the prevalence of the AMDV in American minks was determined to be 22.70%, which could suggest that American mink poses a threat to native species [41]. The geographic distribution of seropositive animals suggests that the virus had spread to all areas that were colonised by European mink. Preliminary results revealed that the AMDV could complicate the European mink protection program in France, but further research is needed to determine the role of the AMDV in the decline of the European mink population [41]. In Spain, by using the serological methods, the prevalence of anti-AMDV antibodies was found to range from 0.00 to 32.40% (A.M. 16.20%) in free-ranging American minks [5,13] and from 0.00 to 32.00% (A.M. 19.00%) in European minks [5,13,38]. The prevalence of the AMDV in free-ranging American minks was studied extensively by Manas et al. [13]. The search for anti-AMDV antibodies lasted 16 years to determine the consequences of AMDV infections for the American mink population. A total of 1735 samples were collected from free-ranging American mink for CIEP analysis. The prevalence of anti-AMDV antibodies was found to be 32.40%. The study demonstrated that AMDV infections were endemic in Spain. In a study that aimed to assess the impact of the AMDV on the Spanish population of European mink, anti-AMDV antibodies were detected in 32.00% of serum samples. The prevalence of antibodies was not significantly correlated with sex, the year of study or body weight. The percentage of anti-AMDV antibodies was very similar in European minks (32.00%) and free-ranging American minks (32.40%) (Table 1). The authors concluded that the AMDV did not contribute to a decrease in the population of either mink species [13]. Gong et al. [37] performed a systematic review and a meta-analysis to assess the seroprevalence of AD in farmed minks in China [37]. They analysed a total of 170569 samples that were detected using CIEP in 31 studies and 4434 samples that were detected using ELISA in seven studies. Based on the data for 1981-2017, the prevalence of anti-AMDV antibodies was found to be between 56.70% (CIEP) and 76.40% (ELISA) (Table 1) [37]. The prevalence of anti-AMDV antibodies in Poland was investigated in the largest population of animals by Zalewski et al. [36]. A total of 1153 feral American mink from nine sites were studied. The animals were screened for AMDV antibodies with AMDV VP2 ELISA according to a procedure that was previously described by Knuuttila [29,30]. The prevalence of anti-AMDV antibodies was determined to be 69.60%, but it varied between regions, sex and age groups and seasons. The prevalence of antibodies ranged from 46.10% in the northern region to 82.60% in the western region [36]. The cited authors also confirmed that AMDV circulation in feral mink can lead to potential spillover to native species, such as pine marten (*Martes martes*), European polecat (*Mustela putorius*), stone marten (*Martes foina*), river otter (*Lutra lutra*) and European badger (*Meles meles*) [36]. This most recent serological study revealed the scale of the problem in the Polish mink population, and further research is needed to confirm the reported results.

## 4. Molecular Epidemiology of the AMDV

Research into the molecular epidemiology of the AMDV mainly relies on analyses of NS1 and VP2 genes [11,12,20,22,24].

The genetic diversity of the AMDV in American minks was investigated in Sweden [21]. A total of 35 samples were analysed, including 31 samples from 15 Swedish farms and four samples from three Finnish farms. In these countries, AD posed a chronic epidemic threat, where it was diagnosed with the use of CIEP. Molecular analyses involved semi-nested PCR with primers that were designed to amplify a 390-nucleotide fragment of the AMDV NS1 gene that was previously described by Bloom et al. [45]. Thirty-five isolates were subjected to phylogenetic analysis, and the results were compared with the sequences of the ADV-G, ADV-K, ADV-SL3, United and Utah-1 variants in the database of the National Center for Biotechnology Information (NCBI). Phylogenetic trees based on deduced amino acid (aa) sequences revealed that 40 AMDVs could be reliably divided into three subgroups. The first subgroup consists mostly of Swedish variants of AMDV from all investigated regions, as well as the highly pathogenic United variant. The second subgroup is a mixture that included the highly pathogenic Danish ADV-K variant, viruses from various Swedish regions and two variants from Finland. The third subgroup includes two samples from a Finnish farm, the nonpathogenic ADV-G and the intermediate ADV-SL3 variant, as well as the highly pathogenic Utah-1 variant. Virulence markers could not be determined at the genomic level because variants that differ considerably in virulence (ADV-G, ADV-SL3 and Utah-1) belong to the third subgroup. In 2004–2009, an epidemiological study of free-ranging American minks was also conducted in Sweden [43]. The aim of the study was to determine the prevalence of anti-AMDV antibodies and the presence of viral DNA in 144 minks. Serum samples that were obtained via hunter harvesting were analysed using a VP2 ELISA test [29,30]. Anti-AMDV antibodies were identified in 46.10% of the samples (Table 1). Spleen/liver samples were analysed using the PCR assay described by Jensen et al. [27]. AMDV DNA was detected in 57.60% of the samples. The authors found that 22.00% of the samples that were examined using ELISA and PCR produced different results, where more than 17.00% of the samples tested positive in the ELISA testing, but negative in the PCR testing. The prevalence of AMDV infection was found to increase with age since the AMDV antibody titres were two-fold higher in two-year-old free-ranging American minks than in juveniles. No relationship was found between AMDV infection in free-ranging minks and liver weight, but in male minks, a significant difference in relative spleen weight was noted between AMDV-positive (spleen weight was higher) and AMDV-negative minks. The phylogenetic analysis of NS1 gene sequences revealed that two of the four genetic AMDV groups were previously described in farmed minks [21,43]. These findings suggest that similar AMDVs exist in farmed and free-ranging minks. Two of the analysed NS1 gene sequences were different, whereas the remaining NS1 gene sequences were similar to Swedish and Danish sequences in 97.50–81.50% of cases [21,40,43]. 

The epidemiology of AD was also studied on Finnish mink farms [12]. The infection was widespread on six commercial farms in the Finnish regions of Ostrobothnia and North Ostrobothnia, where a total of 17 minks were analysed. In the first stage of the analysis, mink sera were examined using CIEP, followed by the semi-nested PCR approach that was previously described by Olofsson [21]. Anti-AMDV antibodies were detected in 76.47% of the samples, while AMDV DNA was detected in 82.35% of the samples (Table 1). One region of the major NS1 gene was amplified, sequenced and subjected to phylogenetic analysis. The identified variants demonstrated 86.00–100.00% nucleotide similarity and were similar between farms. The phylogenetic analysis confirmed the presence of AMDV from three groups, all of which contained Finnish variants. The phylogenetic tree revealed that three lineages of the AMDV were independently introduced to Finland. These findings suggest that these AMDVs were not clustered into genotypes based on geographic origin, year of isolation or pathogenicity. The main conclusion was that the isolates identified on mink farms around the world had originated from North America because American mink were introduced to Europe and Asia for breeding purposes and to restock natural populations. The detailed routes of mink invasion or AD transmission have not been elucidated. The observed changes in the conserved region of the AMDV NS1 gene, which could have contributed to the emergence of more virulent variants, have not been explained [12]. Knuuttila et al. [44] analysed the prevalence, distribution, transmission and diversity of AMDV in Finnish free-ranging mustelids: American minks (*Neogale vison*), European badgers (*Meles meles*), European polecats (*Mustela putorius*), Eurasian otters (*Lutra lutra*), European pine martens (*Martes martes*), least weasels (*Mustela nivalis*), stoats (*Mustela erminea*) and wolverines *(Gulo gulo)* [44]. Anti-AMDV antibodies were detected using AMDV VP2 ELISA [30] and/or AMDV DNA was detected using real-time PCR with new primers corresponding to the AMDV-G NS1 gene [30,44]. A total of 308 samples representing 8 mustelid species and 17 administrative regions were tested in the cited study. Positive samples were detected across Finland in 54.40% (31/57) of feral American minks (*Neogale vison*), 26.90% (7/26) of European badgers (*Meles meles*) and 7.14% (1/14) of European polecats (*Mustela putorius*) (Table 1). The samples from Eurasian otters (*Lutra lutra*), European pine martens (*Martes martes*), least weasels (*Mustela nivalis*), stoats (*Mustela erminea*) and wolverines *(Gulo gulo)* were negative (Table 1) [44]. The prevalences were higher in American minks and badgers. Phylogenetic clusters were not formed based on species, geographic origin or year, excluding four divergent sequences from Estonian badgers, which formed a separate phylogroup that was distinct from the remaining AMDV variants. That study demonstrated that the AMDV was prevalent in certain species of Finnish free-ranging mustelids and widely distributed across Finland. Free-ranging mustelids also carried both variants that were similar to those found by Knuuttila et al. [12,44] in farmed minks, as well as distinct variants that could represent new amdoparvovirus variants [44]. This result was confirmed in 2019 by Virtanen et al. [46]. The authors analysed 52 samples from Finnish fur farms and 45 free-ranging mustelids samples from Finland and Estonia. The study demonstrated that the AMDV is highly diverse in Finland and globally, and it points to extensive virus circulation between countries. Free-ranging and farm variants of the AMDV were mixed in phylogenetic trees, which suggests that the virus was transmitted between farms and the wild, but further research is needed to confirm these observations [46].

In Denmark, nationwide efforts to eradicate the AMDV from mink farms have been made since 1976 due to the severe economic consequences of the disease. Despite the above, around 5.00% of Danish mink farms harboured the virus in 2001 [47]. The diversity of AMDV variants within the NS1 gene was studied in 2004–2005 [47]. AMDV DNA was amplified using PCR with primers based on the protocol developed by Bloom et al. and Olofsson et al. [21,45]. The amplified DNA was subjected to phylogenetic analysis. In the first year of the study, 162 AMDV isolates were obtained from 79 mink farms in the Jutland Peninsula, and in the second year of the study, 112 isolates were obtained from 51 seropositive farms. An additional 50 isolates from 32 infected farms in other Danish regions were collected to fully assess the risk of the AD epidemic. Twenty isolates from eight Dutch farms were also included in the phylogenetic analysis. The AMDV variants from Danish farms were phylogenetically diverse. Many of the identified sequences were also detected in other European countries, including Finland, Sweden and the Netherlands [47]. In 2017, Ryt-Hansen et al. performed a global phylogenetic analysis of the AMDV that was isolated from mink farms [22]. It was the first large-scale phylogenetic study of partial NS1 sequences in the contemporary AMDVs collected from around the world. The study demonstrated that partial NS1 sequencing can be used to identify variants belonging to major clusters and when combined with epidemiological data, it is a helpful tool for tracking outbreaks. The animals were chosen mainly by a veterinarian or a farmer based on clinical signs. Mink carcasses, blood and spleen samples were submitted to the Danish National Veterinary Institute. Samples were collected from a total of 525 animals from 13 countries: Canada, Denmark, Finland, Greece, the Netherlands, Iceland, Italy, Latvia, Lithuania, Poland, Sweden, Spain and the USA. The samples were analysed using CIEP, AMDV-G ELISA [26] and a PCR protocol that was previously described by Jensen [27] with some modifications [23]. The variants from Danish outbreaks in Saeby and Holstebro formed separate clusters in the phylogenetic tree. The AMDVs that were sampled in the USA and Canada clustered tightly, with pairwise sequence similarity of up to 94%. Two sequences originating from mainland Canada were grouped with the sequences obtained during a more recent outbreak in Newfoundland [48]. The Danish Saeby cluster and Finnish sequences were closely related to the North American sequences. Sequences from Lithuania and Sweden were similar in up to 96.00% of cases. The majority of Greek sequences clustered with sequences from the Netherlands, but sequences that originated from the Netherlands, Italy, Sweden and Poland were generally dispersed across several clusters containing a mixture of isolates from different European countries. Spanish sequences clustered closely with 98.00–100.00% of the Finnish isolates. Wild mink sequences from Iceland were most closely related to Swedish sequences from farmed minks, but their homology did not exceed 93.30%. Danish sequences from the Saeby cluster resembled Latvian sequences (with up to 96.00% pairwise sequence similarity). Two Swedish sequences were highly similar (98.00% pairwise identity) to the sequences originating from the 2015 outbreak on the Danish island of Zealand, which lies close to Sweden. The high genetic diversity was attributed to the fact that many of the sampled countries had not implemented clear eradication strategies and had experienced enzootic circulation of the AMDV, which increased the risk of introducing different variants. The authors concluded that full genome sequencing is required to reliably track viral transmission routes across farms [22].

The molecular epidemiology of the AMDV in free-ranging minks, farmed minks and other mammals was also studied in Estonia in 2007–2010 [11]. Samples for analyses were obtained from 51 farmed minks and 152 free-ranging minks, including 27 American minks, 4 European minks (*Mustela lutreola*), 49 pine martens (*Mustela martes*), 42 polecats (*Mustela putorius*), 23 raccoon dogs (*Nyctereutes procyonides*), 4 badgers (*Meles meles*), 2 otters (*Lutra Lutra*) and 1 stone marten (*Mustela foina*). Molecular analyses involved semi-nested PCR with previously described primers [19,21,45]. Positive results were reported in 14.80% (4/27) of free-ranging minks and 21.60% (11/51) of farmed minks (Table 1). Two global phylogenies were built: one based on NS1 (336 bp, 151 taxa from nine countries), and the other based on a combined NS1–VP2 dataset (871 bp, 40 taxa from six countries). The AMDV genotypes did not cluster according to geographic origin, which suggests that farmed minks were transported from multiple sources. Despite the above, one subclade in both phylogenies comprised only isolates from farmed minks, while several subclades comprised only isolates from free-ranging minks, which indicates that some isolates may be more prevalent in the wild, whereas others are more frequently encountered in farmed animals [11].

The first study to describe AMDV sequence variants in Spain was conducted by Manas et al. in 2001 [5]. The authors analysed AMDV sequences that were obtained from free-ranging species: American mink, European mink and Eurasian otter. The main conclusion was that none of the Spanish sequences was identical with sequences that were described previously (ADV-G, ADV-TR, ADV-Utah, ADV-Pullman). However, the sequences from the Eurasian otters were similar to Danish variants and the most pathogenic American variants [5]. In 2020, Prieto et al. characterised 37 AMDV isolates from different types of samples (nest box, cage, slaughter box, spleen, catching gloves) on 17 farms located in Spain, Portugal and France [24]. NS1 and VP2 genes were analysed using the PCR assay described previously by Oie et al. and Olofsson et al. [19,21]. NS1 sequences were characterised by 83.20–100.00% similarity, and VP2 sequences were similar in 91.00–100.00% of cases. AMDV sequences from Spanish farms were divided into three clades based on the global NS1 phylogenetic tree and into four main clades based on the VP2 phylogenetic tree. Based on the global NS1 phylogenetic tree, the distribution of Spanish variants was compared with the variants described in 2017 by Ryt-Hansen et al. [22]. The sequences from clade III were grouped only with the previously described Spanish sequences [22] and they were not closely related to the variants that were identified in other countries. The sequences from clade II clustered mainly with the variants from Poland, Italy, Greece and Sweden. The sequences from clade I appeared fragmented: subclade Ia sequences clustered with the sequences from Poland, the Netherlands and Greece, whereas the sequences from subclades Ib and Ic formed a large cluster with variants from different countries around the world. Based on the global phylogenetic tree for VP2 sequences, the sequences of Spanish clades II and III were quite conserved and were not bound by a close relationship with those reported in other countries. In contrast, clades I and IV sequences were similar to the sequences from other countries. In clade I, the Spanish sequences were most closely related to the sequences from Poland, Finland, Russia and Belarus, whereas the clade IV sequences were grouped mainly with the variants from Finland and Denmark [22,24]. This same research group investigated the applicability of real-time PCR (qPCR) for detecting the sources of AMDV infections in the farm environment [8,9,49]. The faeces of infected mink were characterised using a high viral load, which contributed to environmental contamination. 

For this reason, the farm environment could be an important secondary source of infection. The cited study relied on a commercial qPCR kit that targets the NS1 gene (Genesig Advanced Real-Time PCR Detection Kit for Aleutian Disease Virus, PrimerDesign™ Ltd., Eastleigh, UK), which detects the expression of the non-structural protein I (NS1) gene. The applied kit contains a FAM-labelled Taqman^®^ probe for pathogen detection and a VIC-labelled Taqman^®^ probe for the detection of internal positive control (IPC) DNA. A total of 74 double swabs were collected from various sites on 10 mink farms with different AD epidemiological statuses. All samples from five farms that were considered free of AD were negative. The swabs that were collected from vehicle tyres, shoes, clothing and the staff room on one farm tested positive for AD in qPCR. The study demonstrated that qPCR is an effective technique for detecting AMDV contamination in the farm environment [49]. In 2017, the same research team used the above commercial qPCR kit that targets the NS1 gene to analyse 114 samples from different sites on seven infected mink farms. The prevalence of AMDV DNA was found to be between 69.30 and 81.60%, and it varied across extraction methods [8]. These findings confirmed the authors’ previous observations that positive farms are highly contaminated. These results could be used to improve cleaning and disinfection procedures, as well as biosecurity protocols on farms [8]. In 2018, Prieto et al. [9] relied on the same commercial qPCR kit to identify AMDV-positive *Fannia canicularis*, *Musca domestica* and *Lucilia sericata* flies on a mink farm in north-western Spain where no eradication measures had been applied. Sequence analysis revealed 100.00% homology of all NS1 sequences from flies and environmental samples. This observation suggests that the farm was contaminated with a single AMDV strain and that flies could be used as indicators of AMDV contamination in the farm environment [9].

The first reports describing genetic variants of the AMDV circulating in Polish minks population was published in 2016 [50]. The presence of anti-AMDV antibodies and the polymorphism of the VP2 gene of the AMDV were analysed on two mink farms with breeding stocks of 2000 and 1000 females, respectively, using CIEP and PCR with the use of the methodology described by Costello et al. [51]. Anti-AMDV antibodies were detected in 60.00% of the breeding stock. The isolates from both farms were grouped in a single clade (group I) but occupied two different branches. The sequences from the first branch shared the same clade with Irish isolates and Chinese variants. The sequences from the second branch were grouped only with the sequences isolated in Poland. These differences suggest that AMDV was introduced to Poland at least twice [50]. Jakubczak et al. also examined 20 free-ranging American mink and 11 farmed mink [35]. The presence of the VP2 gene was determined using PCR according to a previously described protocol [45,51]. The AMDV was detected in 35.00% (7/20) of free-ranging American minks and 100.00% of farmed minks (11/11). The phylogenetic analysis revealed considerable similarities between the Polish variants of the AMDV and the variants that were isolated from Ireland and Russia. The authors concluded that the AMDV variants that infected both farmed and free-ranging minks could have originated from a common source but split into two separate subgroups that can be identified based on their distinctive characteristics [35]. In 2017, Siemionek et al. relied on CIEP and PCR assays to assess the epidemiological status of mink farms [52]. PCR was performed based on the protocol that was previously described by Oie et al. [19]. Molecular analyses of 101 spleen samples from seropositive mink confirmed the presence of the AMDV in 10.89% of the animals (11/101). All 11 amplicons were new variants of the AMDV VP2 gene that was isolated from Polish mink farms (acc. no. KT203355-KT203365) [52]. The epidemiological status of 27 mink farms in seven Polish voivodeships was also investigated in 2019 by Kowalczyk et al. [53]. A total of 250 blood, spleen and environmental samples were collected. The virus was detected in animal tissues and the farm environment in a PCR assay based on a fragment of the NS1 gene that was previously described by Jensen et al. [27]. The number of AMDV gene copies was determined with qPCR using a commercial AMDV kit targeting the fragment of the sequence encoding the NS1 gene (Gensig AIDV Advanced Kit PrimerDesign™ Ltd., Eastleigh, UK) [43]. The average viral load reached 10^8^ copies in the spleen, 10^5^ copies in blood and 10^3^ copies in environmental samples [53]. Four main clades were identified in the phylogenetic analysis. The first clade consisted of isolates from northwest Poland, which were characterised by low variation within the group and more than 99.00% similarity with the variants isolated in Greece and the Netherlands in 2016 [22,53]. The second clade contained isolates from eastern Poland with more than 94% similarity to the variants isolated in Lithuania, Sweden and Italy [22,53]. The third clade was composed mainly of isolates from the Wielkopolska and Pomeranian voivodeships, which were characterised by more than 96% similarity with the variants isolated in Poland, the Netherlands and Denmark [22,23,53]. The fourth clade comprised isolates from the Podkarpackie and Małopolska voivodeships with more than 97% homology with the sequences isolated in Poland and Lithuania [22,53]. The results of the phylogenetic analysis indicated that AMDV was highly varied in Poland, probably because the virus had been introduced many times from various sources [53].

In Canada, the first molecular epidemiological study of AMDV was conducted in 2012 in American mink, which is an endemic species in the country [10]. A total of 206 AMDV isolates from free-ranging domesticated minks, hybrids and free-ranging endemic minks in Ontario were analysed. Nucleotide fragments of the AMDV NS1 and VP2 were amplified using PCR with the primers that were previously described by Oie et al. and Olofsson et al. [19,21]. AMDV DNA was detected in 25.00% of free-ranging minks, which points to the presence of an active infection in that population. Amplified NS1 sequences were characterised by 83.00–89.00% similarity to complete sequences of the non-pathogenic cell-culture-adapted ADV-G genome, but the homology between the VP2 sequence and the ADV-G genome was established at 90.00–96.00%. The AMDV isolates from Ontario formed two sub-groups containing NS1 sequences and three sub-groups containing VP2 sequences. These sub-groups were somewhat different, but they were more closely linked with the virus that was circulating in domesticated farmed minks. Molecular analyses revealed that the AMDV had spread from domesticated to free-ranging animals. Clusters were formed by AMDV isolates from free-ranging endemic minks, which are an endogenous reservoir of the virus, as well as isolates from domesticated minks. Biosecurity measures should be observed to prevent the spread of the AMDV between mink farms and the environment [10]. In 2013, Farid relied on CIEP and PCR to detect the AMDV in fur-bearing mammals in Nova Scotia [4]. Between 2009 and 2011, a total of 462 spleen samples were collected from twelve fur-bearing species: American mink (*Neogale vison*) (*N* = 60), short-tailed weasel (*Mustela erminea*) (*N* = 61), fisher (*Martes pennanti*) (*N* = 6), river otter (*Lontra canadensis*) (*N* = 11), coyote (*Canis latrans*) (*N* = 24), red fox (*Vulpes vulpes*) (*N* = 25), raccoon (*Procyon lotor*) (*N* = 85), striped skunk (*Mephitis mephitis*) (*N* = 8), bobcat (*Lynx rufus*) (*N* = 20), muskrat (*Ondatra zibethicus*) (*N* = 59), beaver (*Castor canadensis*) (*N* = 58) and red squirrel (*Tamiasciurus hudsonicus*) (*N* = 45). Anti-AMDV antibodies or AMDV DNA were detected in 93.30% of American minks, 70.50% of short-tailed weasels, 25.00% of striped skunks, 18.20% of river otters, 10.60% of raccoons and 10.00% of bobcats and the remaining samples were negative (Table 1). Such a high number of AMDV-positive samples in various animal species has major epidemiological implications and could pose a serious health threat to animals [4]. A molecular study that investigated the epidemiological status of free-ranging and farmed minks was also conducted in Newfoundland, Canada [48]. The authors analysed a total of 131 NS1 gene sequences from AMDVs circulating between 2004 and 2014 in Canada. NS1 gene sequences were obtained from ten different farms and wild animals (10 American minks, 22 ermines (*Mustela erminea*), 2 Newfoundland lynxes (*Lynx canadensis subsolanus*), 19 red foxes (*Vulpes vulpes deletrix*), 29 Newfoundland pine martens (*Martes americana atrata*), 6 American red squirrels (*Tamiasciurus hudsonicus*) and 40 coyotes (*Canis latrans*)) in Newfoundland (*N* = 97), Nova Scotia (*N* = 13), Ontario (*N* = 9), Wisconsin (*N* = 6) and Denmark (*N* = 6). All the analysed sequences of the AMDV NS1 gene originated from minks because none of the samples from wild species were AMDV-positive [48]. The absence of AMDV infection in wild carnivores other than minks was determined in a small number of animals. NS1 gene sequences formed seven clades containing viruses from different geographic regions, and viruses collected in the same regions formed separate groups. The isolates from Newfoundland and the examined mink farms were characterised by very high levels of genetic diversity. Many animals were simultaneously infected with different variants of AMDV and recombinant strains. The geographic distribution of isolates was only partly described because the identified sequences formed sub-clades that were specific to the analysed countries. The presence of AMDV on farms can promote the emergence of diverse recombinant variants. The transmission of the virus between farms and countries contributes to environmental contamination. The phylogenetic tree derived by Canuti et al. is presented in Figure 1 [48].

In China, the number of mink farms and the population of farmed minks have increased rapidly in recent years. Several species of farmed minks are indigenous, and the remaining species have been imported primarily from Denmark and the USA [12,42]. In China, the epidemiological status of mink farms and the genetic diversity of the AMDV VP2 gene were investigated in 2009–2011 [42]. A total of 18654 serum samples were analysed using CIEP. Anti-AMDV antibodies were detected in 68.67% of the animals. AD was classified as an epidemic disease on Chinese mink farms. A total of 74 spleen samples were collected from seropositive minks on nine farms for a molecular epidemiological study of the AMDV in China. The primers were designed by Qie et al. [19]. AMDV DNA was detected in 79.62% of the samples. The results of the phylogenetic analysis of the AMDV VP2 gene were compared with the VP2 sequences deposited in GenBank (NCBI) in 1970–2009. The Chinese isolates belonged to five independent clades and were characterised by high levels of genetic diversity. More than 50.00% of the Chinese AMDV isolates formed two clades that grouped only Chinese isolates and differed significantly from the sequences that were acquired in other countries. The study demonstrated that both the local AMDVs and imported strains were widespread on Chinese mink farms [42]. Similar results were reported by Wang et al., who analysed 420 samples from farmed minks [20]. Specific serum antibodies were identified using CIEP. Twenty-three samples were also randomly collected from 340 minks for molecular analysis. The PCR protocol and specific primers were designed based on the procedure that was previously described by Oie et al. and Gottschalck et al. [19,54]. The phylogenetic analysis of the VP2 gene involved 23 AMDV isolates from five mink farms. The isolates formed six groups, four of which contained the Chinese isolates. The study also revealed that two viral lines were introduced independently to China. More than 70.00% of the Chinese isolates belonged to two groups. Phylogenetic analyses also demonstrated that the AMDV isolates were not distributed according to geographic origin. The Chinese isolates were ubiquitous in regions with a high number of mink farms. The genetic diversity and phylogenesis of isolates from Chinese mink farms were studied to explore the molecular epidemiology of the AMDV in north-eastern China [55]. In the CIEP assay, seroprevalence on the investigated farms was determined at 41.80%. In the epidemiological study, the prevalence of AMDV NS1 and VP2 genes was analysed with the use of specific primers that were designed according to the method proposed by Oie et al. [19]. Eight new Chinese isolates were identified. AMDV was detected in three provinces in north-eastern China. An analysis of genetic variability in the obtained isolates revealed considerable substitutions in NS1 and VP2 genes, and the substitution rate was highest in the NS1 gene. A phylogenetic analysis of the NS1 gene with a size of 1755 bp and the complete VP2 gene demonstrated that genotype variants were not grouped based on virulence or the geographic location of the isolation site. Local and imported variants of the AMDV were widespread on mink farms in north-eastern China. The described research was the first molecular epidemiological study of AD in north-eastern China that was based on a large fragment of the NS1 gene and the complete VP2 gene. The study provided evidence for new changes in the sequences of AMDV NS1 and VP2 genes. In 2020, the AMDV was isolated from faecal swab samples in China [56]. A total of 291 faecal swabs were obtained from three mink farms in Jilin and Liaoning provinces in north-eastern China. In the group of 291 faecal samples, 157 (54.00%) tested positive for AMDV. Based on previous reports [45,54], a pair of PCR primers was designed to amplify partial nucleotide fragments of the AMDV VP2 gene. A comparison of the nucleotide sequences of the acquired fragments of the VP2 gene revealed that AMDV isolates from mink farms in north-eastern China were closely related to each other but were different from the non-pathogenic AMDV-G strain. Twenty-eight differences were identified in the aa alignment, and they were distributed in all segments of the detected gene. Interestingly, nine aa differences were centrally present in the hypervariable region. In 23 Chinese samples, similar changes were found in fixed positions, especially in the hypervariable regions. These results indicate that Chinese AMDV isolates are highly homologous but different from AMDV from other countries, which is consistent with previous reports [55]. These findings point to the high genetic diversity of the Chinese AMDVs, and they suggest that viral distribution was not linked with geographic origin. Both local and imported AMDV-positive species were prevalent in the Chinese population of farmed minks. The genetic evidence for the AMDV variation and epidemic isolates has important implications for mink-farming practices.

A review of the literature suggests that AMDVs are not clustered into genotypes based on geographic origin, year of isolation or pathogenicity. The main conclusion was that the isolates that were identified on mink farms around the world had originated from North America because American minks were introduced to Europe and Asia for breeding purposes and to restock natural populations. Furthermore, it is concerning that AMDV was identified not only in members of the family Mustelidae but also in other animals, indicating a potential threat to various animal species. The development of molecular biological techniques, such as the polymerase chain reaction (PCR) and sequencing, contributed to significant progress in AD diagnostics. These techniques can not only confirm AMDV infections but also support the accurate identification of the isolates, sources of infection and vectors responsible for the spread of the AMDV between mink farms and the environment. These methods are widely used to detect viral genetic material in the blood, tissues and excretions of infected animals, and they support research into the molecular epidemiology of the AMDV [10,11,12,20,24,44]. This is a very important consideration because other species of free-living animals, in particular those living in the natural environment around American mink farms, may become a reservoir of AMDV. The potential pathogenicity of the AMDV for humans should be investigated, especially among farmworkers and hunters.

## 5. Conclusions

Depending on the applied diagnostic technique, the prevalence of anti-AMDV antibodies or AMDV DNA was found to be 21.60–100.00% in farmed American minks and 0.00–57.60% in free-ranging American minks in Europe, 23.80–70.70% in farmed American minks and 28.60–93.30% in free-ranging American minks in Canada and 56.70% in farmed American minks in China. Anti-AMDV antibodies or AMDV DNA were also detected in other free-living fur-bearing animals in Europe and Canada, where their prevalences were determined to be 0.00–32.00% and 0.00–70.50%, respectively. AD eradication programs have improved the epidemiological status of mink farms in Europe, although new disease foci are still identified. Phylogenetic analyses revealed different levels of similarity between the strains that were isolated from mink farms, but the examined isolates probably originated from a common ancestor. The isolated AMDVs were not grouped based on virulence, time of isolation or the geographic location of mink farms. The epidemiology of the AMDV in populations of free-ranging minks has not been fully elucidated due to the absence of conclusive results. Based on the literature, it can be inferred that the prevalence of AMDV infections is similar in European minks and free-ranging American minks, which could suggest that the AMDV did not contribute to a decrease in the strictly protected population of European minks. The presence of the virus in the environment could undermine the effectiveness of AD eradication programs. New reservoir hosts of the AMDV that are asymptomatic for AD, such as polecats, stone martens, pine martens, common genets, striped skunks and badgers, should be identified to prevent the emergence of new disease foci.

## Figures and Tables

**Figure 1 animals-11-02975-f001:**
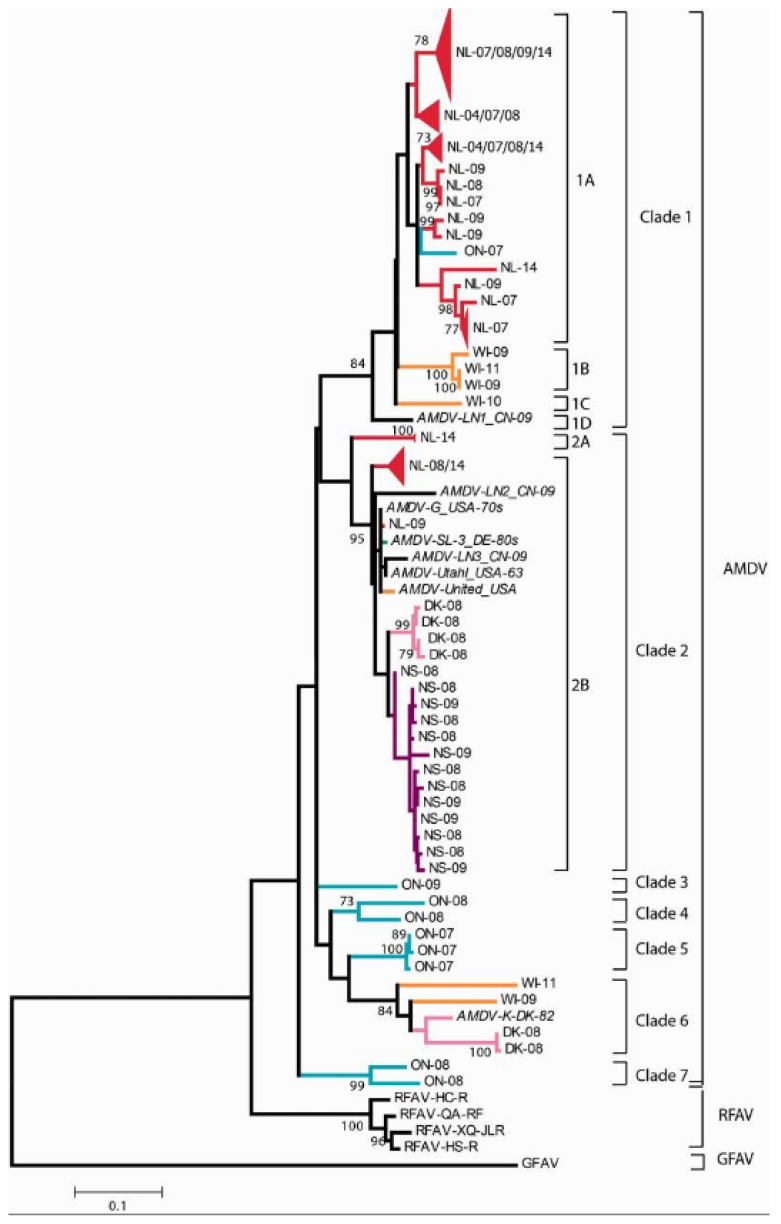
A phylogenetic analysis of partial AMDV NS1 sequences in sequences originating from different areas of the world. The evolutionary history of a fragment of the NS1 region (nt 1207–1690) was inferred using the maximum-likelihood method based on the HKY model (identified as the best-fitting model) in MEGA6. The discrete gamma distribution was used to model the differences in the evolutionary rate across sites (+G = 0.4098 was modelled). The evolutionary history of selected sites was invariable in the rate variation model ((+*I*), 32.514% of sites). The outcome of the bootstrap analysis is shown next to the nodes, and the branch lengths are proportional to the genetic distances indicated by the scale bar. Large groups of sequences originating from the same location and clustered in the same clade were collapsed into a triangular shape at the nodes. The strains are labelled based on the original name (only for the reference sequences, indicated in italics), sampling site (NL: Newfoundland; NS: Nova Scotia; ON: Ontario; WI: Wisconsin; USA: United States of America, state unknown; DK: Denmark; DE: Germany; CN: China) and year. Viral species, clades and subclades are denoted by square brackets. The colours of tree branches correspond to the origins of the samples (red: Newfoundland; purple: Nova Scotia; blue: Ontario; orange: USA; pink: Denmark; green: Germany; black: China). AMDV: Aleutian mink disease virus; RFAV: the raccoon dog and fox amdovirus; GFAV: the gray fox amdoparvovirus [48].

**Table 1 animals-11-02975-t001:** Prevalence of anti-AMDV antibodies and AMDV DNA in American mink, European mink and other species.

Geographic Region	Origin ^1^	Species	No. of Tested Animals	No. of Farms	Prevalence (%)	Detection Methods	Reference
Canada, NS	F	American mink		82	23.80–70.70	CIEP	[3]
	W	American mink	56		28.60	CIEP	[4]
	W	Striped skunks	8		12.50	CIEP	[4]
Canada, NS	W	American mink	60		93.30	PCR or CIEP	[4]
	W	Short-tailed weasel	61		70.50	PCR or CIEP	[4]
	W	Striped skunks	8		25.00	PCR or CIEP	[4]
	W	Otter	11		18.20	PCR or CIEP	[4]
	W	Raccoon	85		10.60	PCR or CIEP	[4]
	W	Bobcat	10		25.00	PCR or CIEP	[4]
	W	Fisher	6		0.00	PCR or CIEP	[4]
	W	Coyote	24		0.00	PCR or CIEP	[4]
	W	Red fox	25		0.00	PCR or CIEP	[4]
	W	Beaver	58		0.00	PCR or CIEP	[4]
	W	Red-squirrel	45		0.00	PCR or CIEP	[4]
	W	Muskrat	59		0.00	PCR or CIEP	[4]
Canada, O	F	Mink spp.		41	46.34	CIEP	[34]
	W	Mink spp.	120		61.66	CIEP	[34]
Canada, O	W	American mink	208		29.00	CIEP	[39]
UK	W	American mink	27		51.85	CIEP	[15]
Denmark, B	W	American mink	142		45.10	CIEP	[40]
Denmark	W	American mink	396		3.30	CIEP	[40]
France	W	American mink	75		22.70	CIEP	[41]
	W	European mink	99		12.20	CIEP	[41]
	W	Polecat	145		11.00	CIEP	[41]
	W	Stone marten	17		23.50	CIEP	[41]
	W	Pine marten	16		6.25	CIEP	[41]
	W	Common genet	68		4.41	CIEP	[41]
Spain	W	American mink	1735		32.40	CIEP	[13]
	W	European mink	492		32.00	CIEP	[13]
	W	American mink	14		0.00	CIEP	[5]
	W	European mink	12		25.00	CIEP	[5]
	W	European mink	84		0.00	CIEP	[38]
China	F	American mink	170,569		56.70	CIEP	[37]
	F	American mink	4434		76.40	ELISA	[37]
	F	American mink	354		35.20	PCR	[37]
	F	American mink	18,654		68.67	CIEP	[42]
Estonia	F	American mink	51	1	21.60	PCR	[11]
	W	American mink	27		14.80	PCR	[11]
	W	European mink	4		0.00	PCR	[11]
	W	Pine marten	49		0.00	PCR	[11]
	W	Polecat	42		0.00	PCR	[11]
	W	Raccoon dog	23		0.00	PCR	[11]
	W	Badger	4		0.00	PCR	[11]
	W	Otter	2		0.00	PCR	[11]
	W	Stone marten	1		0.00	PCR	[11]
Poland	F	American mink	1153		46.10–82.60	ELISA	[36]
	W	American mink	20		35.00	PCR	[35]
	F	American mink	11		100.00	PCR	[35]
Sweden	W	American mink	142		46.10	ELISA	[43]
	W	American mink	139		57.60	PCR	[43]
Finland	F	American mink	17	6	76.47	CIEP	[12]
	F	American mink	17	6	82.35	PCR	[1]
	W	American mink	57		54.40	ELISA and/or PCR	[44]
	W	Badger	26		26.90	ELISA and/or PCR	[44]
	W	Polecat	14		7.14	ELISA and/or PCR	[44]
	W	Otter	24		0.00	ELISA and/or PCR	[44]
	W	Wolverine	1		0.00	ELISA and/or PCR	[44]
	W	Pine marten	183		0.00	ELISA and/or PCR	[44]
	W	Stoat	1		0.00	ELISA and/or PCR	[44]
	W	Least weasel	2		0.00	ELISA and/or PCR	[44]

^1^ W, wild; F, farm; NS, Nova Scotia; O, Ontario; B, Bornholm.

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
