# Peer review of "Seroprevalence and Molecular Epidemiology of Aleutian Disease in Various Countries during 1972–2021: A Review and Meta-Analysis"

_animals, 2021, doi:10.3390/ani11102975_

Round 1
Reviewer 1 Report
Authors prepared the paper reviewing the problem of Aleutian Mink Disease Virus (AMDV). Amdoparvoviruses are very interesting and relatively poorly investigated representatives of Parvoviridae family. AMDV is one of the most serious threat for mink farming. There are no vaccine and therapy that could prevent development of the disease.
Authors prepared the review paper integrating information about prevalence, diagnostics methods and molecular epidemiology. Proposed paper though does not present new information, may be valuable as collect and organize the state of knowledge about AMDV dispersed among different resources.
I have several suggestions and questions.
Simple summary
Line 15 - Rather raccoons, than racconons
Abstract
Authors put details about seroprevalence of AMDV – this is a good idea, as they gather information dispersed among many different papers. Authors included range (from minimum to maximum), I think that it would be worthy to mention about average seroprevalence in particulars regions. Range gives us some knowledge about the scale of epidemiology, but it is wide, and additional data would be very useful.
Introduction
Line 48 – Precise – infected minks
Line 49 – I think that slightly better and more precise would be “excrete viral particles or AMDV particles
Line 53 - Authors write about course of disease and I think that this part should be slightly extended. What is effect and possible outcome of overproduction and deposition of autoimmune complexes ?”
Line 54-56 - This is true, but this part should be moved to the part regarding phylogentics or just removed as the same information is repeated in the further part of the text.
Lines 57-62 - Quite nice part of the text, giving the good background. Authors should consider to move this part to the beginning of introduction.
Line 71 - What means characteristic ? Just hypervariable region.
Line 104 - As I pointed in introduction - please add information about average seroprevalence. You put min and max values, but as you see the range is very wide, therefore, it is worthy to include for example mean values.
Line 122 - Repeated information from line – 106
Line 123 - It wolud be worthy to put in brackets the prevalence of AMDV in particular species.
Line 132 - Add mean values
Line 145 - Gong evaluated seroprevalence not only by CIEP but there were also samples tested by ELISA. I think that it should be mentioned in the text and in the table.
Line 146 – Strains or samples?
Table 1 – Number of ranchers - Ranchers or ranches or farms ?
If you mean farms - complete information for the rest of farm animals - for examples Gong et al. 2020
Molecular epidemiology of AMDV
Line 165 - We have also some research applying NGS technologies, which becoming growing trend - Authors should mention about it. NGS is used both for phylogenetic analysis relation and animal selection. I think that Authors should discuss this issue more comprehensively.
Line 176 - Tree was based on the nucleotide sequence or aa sequence ? It is important as nt sequence=!aa sequence - remember about synonymous mutations. What means 40 AMDVs - sequences, variants, isolates ?
Line 194- Authors mean antibody titer ?
Line 241 - Which farmed minks ? Authors thought about variants from previous Knuuttila study ? Please specify this information.
Line 246 – Circulation instead of exchange.
Line 260 - Please mention about Dutch isolates. Are they formed separate clade ?
Line 438 - Please specify host as "sequences from AMDV circulating" is not precise enough
Line 440 – Include in brackets the number of free-ranging and farmed animals.
Line 441 - It is not clear, as in the previous sentence you write that both the farmed and free-ranging minks were subjected to analysis, and here we have again information about another 10 minks, they were not included as a group with the rest?
Line 452 – Include average prevalence, minimum and maximum.
Line 476 – Please correct – MDV to AMDV
Line 514 - Please correct – faucal to faecal
Line 521 - genes or gene ?
I am not sure if sections are numbered properly – 4. Molecular epidemiology, and after that 6. Conlusions.
Hope that my comments are helpful.
Reviewer 2 Report
L26-27 American mink and European mink. It is two different species but not different regions of the same species.
L46 …members of the Mustelidae family…
L46 Mustelidae family should be in regular style not italic, please check whole text.
L56-57 very similar sentence in Abstract L33-34. The authors should consider replacing L56-67 sentence to results/conclusions sections. Can it be that different genotypes groups by host? I think it should not be the case.
L103-108 Equalize, round once or twice to decimal point. please check whole text.
Table 1. As I see data, when analysing the same study, highest prevalence was determined in American mink and lower virus infections rates were observed in other animals. Could you discuss it and explain reasons for such findings?
L164 how the observed prevalence results can be affected by different target genes or different molecular techniques (for example conventional PCR, semi-nested PCR and so on)? It should be more discussed.
L320 L353 unify. short or long dash
L538-558 or/and at the end of 5 section. Please provide ideas for further research, what studies could resolve the phylogenetic clustering pattern of AMDV strains. What do you propose?
Author Response
Responses to Reviewer 2 Comments
Point 1
L26-27 American mink and European mink. It is two different species but not different regions of the same species.
Response 1:
This is not so obvious. American mink are farmed in America, Europe and Asia, but they are also found in the wild on these continents. In turn, the European mink is endemic to Europe, but it is not farmed. Natural habitats for the European mink have disappeared as a result of environmental pollution. In several countries, such as Spain, France and the UK, attempts are being made to reintroduce this species to the natural environment.
The words “American” and “European” were removed. The modified sentence reads as follows: “This article analyses the seroprevalence of the AMDV infection in American, European mink and other species around the world, and reviews recent knowledge relating to the molecular epidemiology of AMDV”. (in green)
Point 2:
L46 …members of the Mustelidae family…
Response 2: The relevant corrections were made (in green)
Point 3:
L46 Mustelidae family should be in regular style not italic, please check whole text.
Response 3:
The relevant corrections were made (in green)
Point 4:
L56-57 very similar sentence in Abstract L33-34. The authors should consider replacing L56-67 sentence to results/conclusions sections. Can it be that different genotypes groups by host? I think it should not be the case.
Response 4: No, different genotypes are not grouped by host. The relevant information was also provided in the Conclusions section, lines 558-559.
Point 5:
L103-108 Equalize, round once or twice to decimal point. please check whole text.
Response 5: All figures in the text were rounded twice to decimal point (in green).
Point 6:
Table 1. As I see data, when analysing the same study, highest prevalence was determined in American mink and lower virus infections rates were observed in other animals. Could you discuss it and explain reasons for such findings?
Response 6. The highest prevalence of AMDV infection in American mink is considered the main source of infection and the main reservoir of AMDV. In addition, many European countries have not implemented national programmes for controlling AD in farmed American mink. Denmark is an exception, where mandatory AD eradication programmes are in place and where farmed mink have been monitored for several years. However, isolated outbreaks are still reported every few years. According to some researchers, the source of AMDV may be other weasel species occurring in the wild. Systematic research is also being carried out on American mink farms in Canada, but the results have not been published in reputable journals.
The lack of mandatory AMDV testing for imported American mink could contribute to the spread of infections between American mink herds. Fortunately, farms with a high epidemic status purchase mink from AD-free farms; therefore, not all purchased mink necessarily have AMDV-specific antibodies.
Lower testing rates in other species may be due to several reasons. Firstly, the mortality rates associated with AMDV infection are unknown, and most studies involve live animals. Secondly, little is known about the pathogenesis of AMDV infection in other species and the mechanisms of immunity. For example, several studies have been conducted on ferrets infected with a mutant variant of mink AMDV. It is now thought that the Aleutian ferret disease virus causes asymptomatic infections and abnormalities, including in kidney function, that can lead to death in the long term. Thirdly, samples are collected randomly; therefore, they are difficult to collect according to a research plan across large areas and within a defined period. Most often, the number of samples is small, and larger numbers of samples collected over a longer period have been analysed only in a limited number of studies.
AMD control programmes are increasingly implemented by European fur-bearing organisations to reduce the prevalence of AMDV in farms.
Point 7:
L164 how the observed prevalence results can be affected by different target genes or different molecular techniques (for example conventional PCR, semi-nested PCR and so on)? It should be more discussed.
Response 7: We agree with the Reviewer. The impact of various factors on the observed prevalence results had been discussed at the beginning of the original manuscript. However, the article was too long and had to be abridged. We are planning to write a review article about AD diagnostics, including VP2 ELISA, AMDV-G ELISA (ELISA Danad antigen, Kopenhagen Fur, Glostrup, Denmark), peptide ELISA, VP2332-452 ELISA, and antigen-capture (AC) ELISA, in the future. Molecular methods such as conventional and real-time PCR, SYBR green, EvaGreen and molecular probes, NGS and WGS will also be discussed.
Point 8:
L320 L353 unify. short or long dash
Response 8:
A short dash was used in both instances (in green).
Point 9:
L538-558 or/and at the end of 5 section. Please provide ideas for further research, what studies could resolve the phylogenetic clustering pattern of AMDV strains. What do you propose?
Response 9. The directions for future research were described at the end of the paragraph: ”The development of molecular biological techniques, such as the polymerase chain reaction (PCR) and sequencing, have contributed to significant progress in AD diagnostics. These techniques not only confirm an AMDV infection, but also support accurate identification of isolates, sources of infection, and vectors responsible for the spread of AMDV between mink farms and the environment. These methods are widely used to detect viral genetic material in the blood, tissues and excretions of infected animals, and they support research into the molecular epidemiology of AMDV [10-12,20,24,44]. This is a very important consideration because other species of free-living animals, in particular those living in the natural environment around American mink farms, may become a reservoir of AMDV. The potential pathogenicity of AMDV for humans should be investigated, especially among farmworkers and hunters.”

Reviewer 3 Report
This is a very nice review article which details AMDV- a largely understudied and under reported area. It is well researched and generally well written. The molecular epidemiology bit is interesting, but seems never ending, so maybe breaking this up into sub sections with country subheadings.
I have some more minor comments below, but this doesn’t detract from the fact that this is a very interesting manuscript. Most of these are just ideas and suggestions, and I will leave it up to the authors discretion.
Line 18-21- I would reword this as it doesn’t make complete sense
Line 25- maybe reword to … for other members of the Mustelidea family…
Line 29 and in several other places- why do you group antibodies and DNA together, these are quite different things?
Line 46- maybe reword to … for other members of the Mustelidea family…
Line 47- is this raccoons?
Line 48-49- maybe faeces containing AMDV?
Line 82- would insertion of the prisma statement here be useful?
Line 87-88- why were these countries chosen? Are these the only ones where studies have been done? If so, maybe worth saying that
Line 101- maybe this is meant to be techniques?
Line 104-106- the countries for these prevalence’s would be useful
Line 117- a prevalence for these animals would be nice
Line 122- 123- this can be deleted and replaced with (Table 1)
Line 123-125- again a % with these non mink animals would be nice if possible
Line 131- comma after methods
Line 136- a mention of how the longitudinal data differed in here may be nice
Line 156- maybe a mention of the specific spillover species could be useful?
Line 190- and other places- you mention PCR assays. It would be nice to mention the target gene for these assays where appropriate so the reader can begin to work out how good that test would be
Line 199- you mention three groups before- is this meant to be 4?
Line 202- this is a bit unclear- how can it be identical and yet 97-81%?
Line 227- maybe mention the gene in here?
Line 234- which animal species? And is it risk or prevalence?
Line 242- does this mean that there is a risk of introduction into a mink farm from wild mustelids?
Line 252- you mention the stability of the NS1 gene- is this correct? Perhaps just needs rewording, but I could be wrong
Line 253- and other places- you do not isolate by PCR- you amplify by PCR
Line 260-261- please reword this as it doesn’t make sense
Line 279- maybe reword this- cant be homologous to 96% as homology means the same?
Line 283- again, 98-100% of Finnish isolates. Could you be more specific- its either them all or it isn’t
Line 311- is this three in total, or more?
Line 324-330- is it possible to mention similarity levels in here?
Line 334-336- an image would be very useful here I think
Line 346- probably statuses here
Line 348- tyres
Line 349- is it controlling or just detecting the virus?
Line 369- again a similarity score in here may be useful
Line 384- not completely sure that this makes total sense to me
Line 393- encoding the NS1 gene (Reword)
Line 398- similarity and not homology
Line 434- a % in the other animals would be nice, or just a comment that they were all negative
Line 512-513- this seems a bit weird as you don’t get aa changes in a gene?
Line 520-522 this seems a bit weird as you don’t get aa changes in a gene?
Line 247- similarity rather than homology
Line 556- asymptomatic?
Author Response
Responses to Reviewer 3 Comments
Point 1: Line 18-21- I would reword this as it doesn’t make complete sense
Response 1: The sentence “Aleutian mink disease virus circulating between farmed, free-living mink, environment and could pose a serious threat to the survival of native species” was deleted (in blue).
Point 2: Line 25- maybe reword to … for other members of the Mustelidea family…
Response 2: The suggested correction was made: “The disease is caused by the Aleutian mink disease virus (AMDV) which also poses a health risk for other members of the Mustelidae family, including wild mink, weasels, badgers and other animal species.” (in blue)
Point 3: Line 29 and in several other places- why do you group antibodies and DNA together, these are quite different things?
Response 3: We are aware that these are quite different things, but the aim of the study was to analyse AMDV antibodies or DNA (in blue).
Point 4: Line 46- maybe reword to … for other members of the Mustelidea family…
Response 4: The suggested correction was made: “The disease is caused by the Aleutian mink disease virus (AMDV) which also poses a health risk for other members of the Mustelidae family, including wild mink, weasels, badgers and other animal species such as skunks, raccoons, dogs, cats and mice, as well as humans” (in blue).
Point 5: Line 47- is this raccoons?
Response 5: We apologise for this obvious spelling mistake (in blue).
Point 6: Line 48-49- maybe faeces containing AMDV?
Response 6: As suggested by Reviewer 1, the sentence was modified as follows: “Infected minks with and without symptoms of the disease excrete viral particles with faeces into the environment” (in blue).
Point 7: Line 82- would insertion of the prisma statement here be useful?
Response 7: It would be very useful, and we are planning to do it in our next review article (in blue).
Point 8: Line 87-88- why were these countries chosen? Are these the only ones where studies have been done? If so, maybe worth saying that
Response 8: These countries were chosen because mink farming is or had been an important part of their agricultural production. The sentence was modified accordingly: “These countries were chosen because mink farming is or had been an important part of their agricultural production.” (in blue)
Point 9: Line 101- maybe this is meant to be techniques?
Response 9: The relevant correction was made (in blue).
Point 10: Line 104-106- the countries for these prevalence’s would be useful
Response 10: Prevalence was shown for each group, i.e. free-ranging American mink, farm mink and European mink. The prevalence in the surveyed countries was described below in the text (in blue).
Point 11: Line 117- a prevalence for these animals would be nice
Response 11: This sentence refers only to the risk for the discussed species (in blue).
Point 12: Line 122- 123- this can be deleted and replaced with (Table 1)
Response 12: The relevant corrections were made (in blue).
Point 13: Line 123-125- again a % with these non mink animals would be nice if possible
Response 13: The sentence was modified as follows: “Anti-AMDV antibodies were also identified in 11.00% of polecats (Mustela putorius), 23.50% of stone martens (Martes foina), 6.25% of pine martens (Martes martes) and 4.41% of common genets (Genetta genetta)” (in blue).
Point 14: Line 131- comma after methods
Response 14: The relevant correction was made (in blue).
Point 15: Line 136- a mention of how the longitudinal data differed in here may be nice
Response 15: Differences in longitudinal data will be discussed in our next article (in blue).
Point 16: Line 156- maybe a mention of the specific spillover species could be useful?
Response 16: The sentence was modified as follows: “The cited authors also found that AMDV circulation in feral mink can lead to potential spillover to native species such as pine marten (Martes martes), European polecat (Mustela putorius), stone marten (Martes foina), river otter (Lutra lutra) and European badger (Meles meles)” (in blue).
Point 17: Line 190- and other places- you mention PCR assays. It would be nice to mention the target gene for these assays where appropriate so the reader can begin to work out how good that test would be
Response 17: The virus was detected in animal tissues and the farm environment in a PCR assay based on a fragment of the NS1gene that had been previously described by Jensen et al. (in blue).
The original manuscript had a long section on the diagnostics of AD at the beginning, but it had to be removed because the article was too long. We are planning to write a review about AD diagnostics in the future.
Point 18: Line 199- you mention three groups before- is this meant to be 4?
Response 18: The first part was described by Olofsson, A.; Mittelholzer, C.; Treiberg Berndtsson, L.; Lind, L.; Mejerland, T.; Belák, S. Unusual, high genetic diversity of Aleutian mink disease virus. J. Clin. Microbiol. 1999, 37, 4145-4149, and the second – by Persson, S.; Jensen, T.H.; Blomström, A.L.; Tjernström Appelberg, M.; Magnusson, U. Aleutian mink disease virus in free-ranging mink from Sweden. PLoS One 2015, 10, 1-11. (in blue)
Point 19: Line 202- this is a bit unclear- how can it be identical and yet 97-81%?
Response 19: The word “identical” was replaced with “similar” (in blue).
Point 20: Line 227- maybe mention the gene in here?
Response 20: The gene was named, and the second part of the sentence was deleted: “Anti-AMDV antibodies were detected by AMDV VP2 ELISA [29], and/or AMDV DNA was detected by real-time PCR with new primers corresponding to AMDV-G NS1 gene nt 1662-1684 and 2302-2281 regions of the AMDV-G sequence (GenBank accession number (Acc. No.) JN0404340)” (in blue).
Point 21: Line 234- which animal species? And is it risk or prevalence?
Response 21: The following sentence was deleted: “The animal species was the major risk factor for infection.” The word “risk” was replaced with “prevalence”(in blue).
Point 22: Line 242- does this mean that there is a risk of introduction into a mink farm from wild mustelids?
Response 22: Yes, especially on farms that are not completely isolated from the environment. Farms that serve tasty and wholesome feed mixes in mink cages are invaded by wild mink and other animal species several times a day. The served feed contains 95% of animal or fish by-products, extruded wheat and vitamin supplements. Such farms are excellent sources of wholesome and easily available food. The difficulty in eradicating AMDV on mink farms could suggest that AMDV is carried not only by mink, but also by other animal species living in the wild around mink farms. Theoretically, other animal species should not be able to enter mink farms that abide by strict biosecurity principles (such as double fences with a height of 2 m). One fence must have solid walls, and the other can be a mesh fence. However, many researchers have argued that other reservoirs of AMDV need to be identified. Meaningful results based on a large number of samples from free-living species were obtained only in several studies (in blue).
Point 23: Line 252- you mention the stability of the NS1 gene- is this correct? Perhaps just needs rewording, but I could be wrong
Response 23: The words “and stability” were deleted (in blue).
Point 24: Line 253- and other places- you do not isolate by PCR- you amplify by PCR
Response 24: The relevant corrections were made in the entire manuscript (in blue).
Point 25: Line 260-261- please reword this as it doesn’t make sense
Response 25: The sentence: “The amplified isolated sequences formed three clades grouping only Danish isolates.” was deleted (in blue).
Point 26: Line 279- maybe reword this- cant be homologous to 96% as homology means the same?
Response 26: The word “homologous” was replaced with “similar” (in blue).
Point 27: Line 283- again, 98-100% of Finnish isolates. Could you be more specific- its either them all or it isn’t
Response 27: The phrase “but sequence identity was determined at 93.00%. “ was deleted (in blue).
Point 28: Line 311- is this three in total, or more?
Response 28: Three in total. We apologise for this mistake (in blue).
Point 29: Line 324-330- is it possible to mention similarity levels in here?
Response 29: In our opinion, similarity levels do not have to be mentioned here (in blue).
Point 30: Line 334-336- an image would be very useful here I think
Response 30: It could be useful, but in our opinion, it is not necessary (in blue).
Point 31: Line 346- probably statuses here
Response 31: The relevant correction was made (in blue).
Point 32: Line 348- tyres
Response 32: The relevant correction was made (in blue).
Point 33: Line 349- is it controlling or just detecting the virus?
Response 33: Detecting (in blue).
Point 34: Line 369- again a similarity score in here may be useful
Response 34: It could be useful, but in our opinion, it is not necessary (in blue).
Point 35: Line 384- not completely sure that this makes total sense to me
Response 35: The sentence was deleted (in blue).
Point 36: Line 393- encoding the NS1 gene (Reword)
Response 36: The relevant correction was made (in blue).
Point 37: Line 398- similarity and not homology
Response 37: The word “homology” was replaced with ”similarity” (in blue).
Point 38: Line 434- a % in the other animals would be nice, or just a comment that they were all negative
Response 38: The sentence was modified as follows: “Anti-AMDV antibodies or AMDV DNA was detected in 93.30% of American mink, 70.50% of short-tailed weasels, 25.00% of striped skunks, 18.20% of river otters, 10.60% of raccoons and 10.00% of bobcats, and the remaining samples were negative” (in blue).
Point 39: Line 512-513- this seems a bit weird as you don’t get aa changes in a gene?
Response 39: The sentence was modified as follows: “The study provided evidence for new changes in aa the sequences of AMDV NS1 and VP2 genes.” (in blue)
Point 40: Line 520-522 this seems a bit weird as you don’t get aa changes in a gene?
Response 40: “aa” was removed (in blue).
Point 41: Line 247- similarity rather than homology
Response 41: The word “homology” was replaced with “similarity” (in blue).
Point 42: Line 556- asymptomatic?
Response 42: The relevant correction was made (in blue.
Once again, we would like to thank the Reviewer for valuable comments and suggestions.

Round 2
Reviewer 1 Report
Generally, Authors addressed to most of my comments. In my opinion, paper in present form is sufficiently improved and meets requirements of publication.
Paper organizes information about AMDV in a quite good way and may be a good introduction to the topic of AMDV epidemiology.
I have only minor suggestions.
In some places just “AMDV” is not sufficient, please precise, do you mean variant or strain or isolate? – Line 548
Line 461 – minor issue, but it would be worthy to keep one notation, therefore, I propose put “2” instead of two and “6” instead of six.
Author Response
We would like once again thanks the Reviewer for valuable comments
Point 1
In some places just “AMDV” is not sufficient, please precise, do you mean variant or strain or isolate? – Line 548
Response 1: The relevant correction was made “isolates” (in red).
Point 2
Line 461 – minor issue, but it would be worthy to keep one notation, therefore, I propose put “2” instead of two and “6” instead of six.
Response 2: The relevant correction was made